# Are We Still Mediterranean? Dietary Quality and Adherence in Sicilian Women Undergoing ART: A Prospective Observational Cohort Study

**DOI:** 10.3390/medicina62010023

**Published:** 2025-12-23

**Authors:** Annalisa Liprino, Veronica Corsetti, Filippo Giacone, Giorgio Ivan Russo, Maria Giovanna Asmundo, Sandrine Chamayou, Antonino Guglielmino

**Affiliations:** 1Centro HERA-Unità di Medicina della Riproduzione, Via Barriera del Bosco, 51/53, Sant’Agata li Battiati, 95030 Catania, Italy; filippogiacone@yahoo.it (F.G.); s.chamayou@yahoo.fr (S.C.); angugl2017@gmail.com (A.G.); 2Institute of Translational Pharmacology (IFT)-CNR, Via Fosso del Cavaliere 100, 00133 Rome, Italy; veronicacorsetti@me.com; 3Urology Section, Department of Surgery, University of Catania, 95123 Catania, Italy; giorgioivan1987@gmail.com (G.I.R.); mariagiovannaroberta.asmundo@you.unipa.it (M.G.A.)

**Keywords:** Mediterranean diet, dietary adherence, assisted reproductive technology, infertility, ovarian response, reproductive outcomes, MEDAS score

## Abstract

*Background and Objectives:* The Mediterranean diet is traditionally linked to metabolic balance and improved reproductive health. However, dietary patterns in Mediterranean regions have progressively shifted toward more Westernized models, particularly among women of reproductive age, raising concerns about declining adherence to this historically protective diet. Objective: To assess adherence to the Mediterranean diet among women undergoing assisted reproductive technology (ART) and to explore possible associations with ovarian response and clinical outcomes. *Materials and Methods:* This prospective observational cohort study was conducted at a reproductive clinic in Sicily between 1 June and 31 July 2022. One hundred women aged 18–40 years undergoing infertility assessment and scheduled for controlled ovarian stimulation were enrolled. Mediterranean diet adherence was evaluated using the validated 14-item MEDAS questionnaire during the first clinical visit. ART-related outcomes, including ovarian response and pregnancy rates, were extracted from medical records. *Results:* The mean MEDAS score was 7.6 ± 1.2: 93% of women showed moderate adherence, 3% high adherence, and 4% low adherence. No significant associations were found between MEDAS score, and total oocytes retrieved, MII oocytes, or clinical pregnancy. *Conclusions:* Despite living in a traditionally Mediterranean area, participants demonstrated only moderate adherence to the Mediterranean diet. Although no associations with single-cycle ART outcomes emerged, the findings underscore the need for structured nutritional counseling to reinforce sustained adherence and support long-term reproductive health.

## 1. Introduction

The Mediterranean diet has long been recognized as one of the healthiest dietary patterns worldwide, characterized by a nutritional profile rich in plant-based foods, extra virgin olive oil, whole grains, legumes, fish, and a low intake of red and processed meats [1]. Beyond its nutrient composition, the Mediterranean diet represents a holistic model that incorporates culinary tradition, shared meal preparation, intergenerational transmission of knowledge, and seasonal and territorial selection of food ingredients. In recent years, however, this traditional dietary paradigm has progressively eroded, reflecting broader societal changes in food accessibility, meal timing, and lifestyle rhythms that challenge adherence even within Mediterranean regions.

Infertility affects an estimated 15% of couples worldwide, and its prevalence has steadily increased over the last two decades, partly due to delayed childbearing, environmental exposures, and lifestyle changes [2]. Approximately 35–40% of infertility cases are primarily female in origin, 30–35% are male-associated, and 20–25% are combined or unexplained. The rise in unexplained infertility has prompted growing interest in factors influencing reproductive competence beyond identifiable anatomical and endocrinological causes [3]. Infertility carries significant psychological consequences, frequently manifesting as emotional distress, altered self-identity, and a diminished quality of life for affected individuals and couples [4]. This multidimensional burden underscores the need to consider modifiable lifestyle factors as part of a comprehensive fertility-care framework.

Although procedural advances in ART have steadily enhanced success rates, growing attention is being placed on lifestyle factors—such as dietary habits, physical activity, smoking cessation, sleep quality, and body weight management—which can significantly influence reproductive outcomes [5,6]. In this context, nutrition plays a particularly central role: dietary patterns influence systemic inflammation, mitochondrial function, insulin sensitivity, lipid metabolism, and oxidative stress, all of which are implicated in ovarian physiology, follicular microenvironment quality, and endometrial receptivity [7]. A dietary model rich in monounsaturated fatty acids, polyphenols, carotenoids, folate, and omega-3 fatty acids, such as the Mediterranean diet, may therefore support both oocyte competence and embryo implantation potential [8]. Emerging evidence also implicates dietary patterns in shaping the gut and vaginal microbiota, thereby influencing immune tolerance, metabolic regulation, and early reproductive processes.

However, the cultural and nutritional landscape of Mediterranean countries has undergone a profound shift in recent decades [1]. Changes in food availability, working rhythms, and social meal structures have contributed to a progressive westernization of dietary patterns, even in regions historically linked to the Mediterranean diet [1]. This transition has been particularly evident among younger adults, including women of reproductive age, who increasingly adopt fragmented eating behaviors and greater consumption of processed foods. Such changes raise the question of whether contemporary Mediterranean populations can still be considered adherent to the dietary model that historically characterized their lifestyle and supported beneficial health outcomes.

To accurately assess adherence to the Mediterranean diet in clinical research and practice, the 14-item Mediterranean Diet Adherence Screener (MEDAS), developed and validated in the PREDIMED Trial, has become a widely adopted instrument due to its brevity, reproducibility, and reliable scoring system [9]. It allows stratification into low, moderate, and high adherence levels and is therefore suitable for identifying meaningful nutritional targets within fertility care [10]. Its simplicity, however—while practical—may not fully capture the complexity of contemporary dietary behaviors, a limitation that is increasingly discussed in nutritional epidemiology.

The aim of this study was to evaluate adherence to the Mediterranean diet in a cohort of women aged 18–40 years undergoing infertility treatment at a fertility clinic in Sicily and to describe contemporary dietary behavior in a population for whom nutrition represents a clinically modifiable factor with potential implications for ART outcomes [11]. By situating this investigation within a traditionally Mediterranean region, the study also offers insight into the evolving nutritional landscape of reproductive-age women in Italy, illuminating both clinical and cultural dimensions of diet-related reproductive health. The primary objective of this study was to describe the distribution of MEDAS scores in women undergoing infertility assessment and ART in a Sicilian reproductive medicine center. Secondary exploratory objectives included evaluating associations between MEDAS and BMI, ovarian response, and clinical pregnancy within a single ART cycle.

## 2. Materials and Methods

### 2.1. Study Design and Participants

This was a prospective observational cohort study conducted at the UMR-HERA Reproductive Medicine Center in Sicily, Italy, between 1 June and 31 July 2022. The center is a reproduction clinic providing both infertility diagnostics and assisted reproductive technology (ART) procedures. The primary outcome was the distribution of MEDAS scores. Secondary outcomes included associations between MEDAS score and BMI, oocyte yield, MII oocytes, and clinical pregnancy. All consecutive eligible women aged 18–40 years attending the clinic between 1 June and 31 July 2022 were invited to participate. No formal sample size calculation was performed. Exclusion criteria included: uncontrolled endocrine or metabolic disorders (e.g., untreated thyroid disease, poorly controlled diabetes), significant systemic illness, inability to complete the dietary questionnaire, or missing clinical data. Women with PCOS, endometriosis, or diminished ovarian reserve were not excluded, as these diagnoses represent common infertility conditions and reflect real-world ART populations. Within the female-factor subgroup, the most frequent diagnoses included diminished ovarian reserve, endometriosis, and PCOS. However, detailed distribution was not uniformly available and is therefore not reported in stratified form. A total of 100 women met the inclusion criteria and were enrolled in the study. Height and weight were measured during clinical evaluation, and body mass index (BMI) was calculated as weight (kg)/height (m^2^) [12]. Only first or first monitored stimulation cycles during the study period were included. Women undergoing a repeated cycle within the study timeframe were not enrolled. Reproductive and ART-related outcomes, including total oocytes retrieved, number of metaphase II (MII) oocytes, and clinical pregnancy, were obtained from electronic medical records. Baseline demographic and clinical characteristics are presented in Table 1 and Table 2.

### 2.2. ART Procedures

All participants underwent controlled ovarian stimulation following a short GnRH antagonist protocol, with gonadotropin dosage and trigger strategy individualized based on baseline ovarian reserve parameters and follicular response monitoring. Final oocyte maturation was induced using U-hCG (Gonasi^®^, IBSA, Lugano, Switzerland) or GnRH agonist triptorelin (Fertipeptil, Ferring Pharmaceuticals, St-Prex, Switzerland). Trigger choice (hCG or GnRH agonist) was based on follicular response and the estimated risk of ovarian hyperstimulation syndrome. All patients received standardized luteal phase support following oocyte retrieval. No freeze-all cycles were performed during the study period. Clinicians were blinded to MEDAS results, which were not considered in stimulation planning or clinical decision-making. Clinical pregnancy was defined as the presence of an intrauterine gestational sac with fetal heartbeat visualized at transvaginal ultrasound at 6–7 weeks of gestation.

### 2.3. Dietary Assessment and MEDAS Questionnaire

Adherence to the Mediterranean diet was evaluated using the 14-item Mediterranean Diet Adherence Screener (MEDAS), originally validated in the PREDIMED trial for clinical and research use. The questionnaire (Table 3) was administered prospectively, dietary assessment was performed before ovarian stimulation and prior to protocol planning, during the initial clinical evaluation. Each affirmative response contributing to the Mediterranean pattern scored 1 point (range: 0–14). Overall adherence was categorized as: Low: 0–5; Moderate: 6–9; High: ≥10.

### 2.4. Statistical Analysis

Continuous variables were reported as mean ± SD or median (IQR), as appropriate. Continuous outcomes were compared across MEDAS tertiles using the Kruskal–Wallis test, and categorical outcomes using the χ^2^ test. Associations between the continuous MEDAS score and ART outcomes were analyzed with Spearman’s rank correlation. Analyses were performed using GraphPad Prism (version 10.0; GraphPad Software, San Diego, CA, USA). Statistical significance was set at *p* < 0.05.

### 2.5. Ethical Approval

The study was conducted in accordance with the Declaration of Helsinki. The study protocol, including the use of an online pseudonymized dietary survey, was reviewed and approved by the Institutional Review Board of Centro HERA-Unità di Medicina della Riproduzione (Approval Code: 012022; Approval Date: 17 January 2022).

## 3. Results

Baseline demographic and clinical characteristics according to infertility etiology are presented in Table 1 and Table 2. The mean age of the population was 35.7 ± 3.8 years (median = 36; IQR ≈ 33–38). The mean BMI was 24.1 ± 4.7 kg/m^2^, reflecting a mixed metabolic profile: 56% were of normal weight, 23% overweight, 14% obese, and 7% underweight. The mean MEDAS score was 7.6 ± 1.2. Most participants (93%) showed moderate adherence to the Mediterranean diet, while 3% achieved high adherence and 4% scored in the low range. The score distribution was concentrated within the moderate range (Figure 1). No significant correlations were observed between the MEDAS score, and the total number of oocytes retrieved (*p* = 0.16), number of MII oocytes (*p* = 0.07), or clinical pregnancy (*p* = 0.44). A weak but statistically significant inverse correlation was found between MEDAS score and BMI (Spearman’s r = −0.21, *p* = 0.035), suggesting that greater adherence to the Mediterranean diet was associated with a lower body mass index. No significant association was observed between MEDAS score and AMH levels (Spearman’s r = 0.04, *p* = 0.71). Women who achieved pregnancy had a mean MEDAS score of 7.46 compared with 7.64 among non-pregnant women, with no statistically significant difference (*p* = 0.44). In a logistic regression model, the MEDAS score was not a significant predictor of clinical pregnancy (OR = 0.87; 95% CI 0.62–1.23; *p* = 0.44).

## 4. Discussion

This study shows that women undergoing infertility treatment in our center exhibit moderate, rather than high, adherence to the Mediterranean diet, despite living in a region traditionally associated with this nutritional model. This finding reflects a broader cultural shift toward more Westernized dietary behaviors, a trend documented across Mediterranean countries where ultra-processed foods and fast-paced lifestyles increasingly challenge traditional eating habits, especially among younger adults. In previous studies, beneficial reproductive effects were generally observed among women with high-level adherence. In our cohort, the clustering around moderate adherence may have limited the ability to detect meaningful associations, supporting the hypothesis that metabolic and reproductive benefits may emerge only beyond a certain adherence threshold [13].

The study was conducted during the summer months, when the availability of fresh produce is greater; this may have slightly elevated MEDAS scores and reduced variability across participants. Moreover, the Mediterranean diet likely exerts its reproductive effects through mechanisms involving oxidative stress, inflammatory signaling, and insulin sensitivity, which require high and sustained adherence over time [14,15]. The short recruitment period and convenience sampling may limit generalizability, and seasonal dietary patterns could have influenced MEDAS scores. In this cohort, adherence was largely moderate, and ART outcomes were assessed over a single stimulation cycle, limiting the ability to observe the influence of long-term dietary patterns on oocyte competence or endometrial receptivity. The mixed BMI distribution and the absence of severe metabolic impairment may have further attenuated detectable associations.

Dietary and lifestyle data were self-reported, with potential recall and social desirability bias. The observational design precludes causal inference, and evaluating outcomes within a single ART cycle does not capture cumulative live birth potential. The narrow adherence range, dominated by moderate scores, also limited contrast between groups. Future studies with broader adherence distribution, longer follow-up, and integration of metabolic and microbiome parameters may better elucidate the pathways through which Mediterranean diet patterns influence reproductive health. Notably, structured nutritional counseling was not part of routine care in this cohort, highlighting a clinical opportunity to incorporate individualized and sustained dietary guidance into infertility pathways.

Emerging evidence indicates that dietary patterns influence not only metabolic and reproductive health but may also exert transgenerational effects through epigenetic modulation of gametes and early embryonic development [16,17]. Within the context of infertility care, this expands the meaning of nutrition beyond the individual: diet becomes part of the physiological environment in which reproductive potential is shaped, and future health trajectories may begin [18].

Although no association was observed between MEDAS adherence and ART outcomes within a single stimulation cycle, these results highlight a critical opportunity. Nutritional counseling should be integrated systematically into fertility pathways, not as an optional adjunct, but as a foundation of preconception care—a way to restore the link between women, their bodies, their food traditions, and the health of the generations to come.

From a clinical perspective, structured models of nutritional intervention could include preconception dietary counseling based on Mediterranean principles, integration of nutrition professionals within ART teams, periodic dietary assessment throughout treatment cycles, and combined metabolic–nutritional programs targeting insulin sensitivity, inflammation, and body composition. Such approaches may promote sustained adherence and support long-term reproductive health.

## 5. Conclusions

Women undergoing infertility treatment in this Mediterranean region demonstrated predominantly moderate adherence to the Mediterranean diet. A modest inverse association was observed between MEDAS score and BMI, whereas no relationship emerged with AMH or with ART outcomes within a single stimulation cycle. These findings highlight the need to integrate structured nutritional counseling into fertility care to promote higher and sustained adherence.

Future studies should evaluate long-term dietary adherence and its potential impact across multiple ART cycles, ideally incorporating metabolic, microbiome, and lifestyle data to further clarify the mechanisms through which nutrition may influence reproductive health.

## Figures and Tables

**Figure 1 medicina-62-00023-f001:**
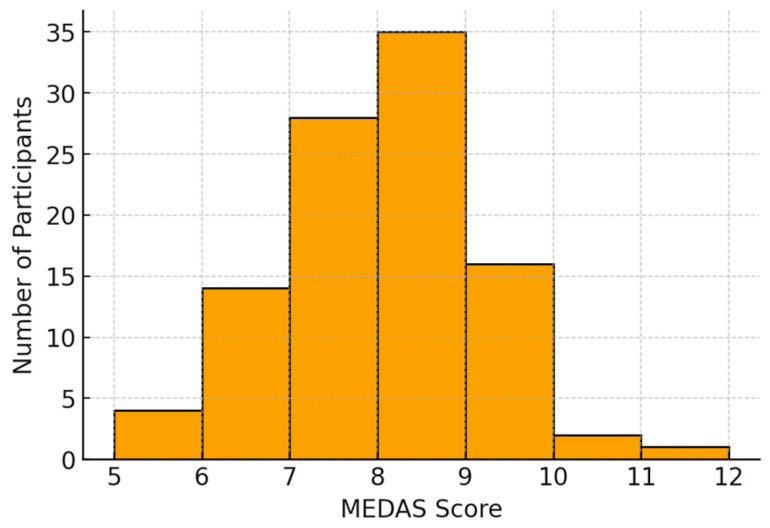
Distribution of Mediterranean diet adherence scores (MEDAS) among women undergoing infertility treatment (*n* = 100). The histogram illustrates the frequency distribution of MEDAS scores, while the superimposed density curve highlights the overall trend of moderate adherence within the study population.

**Table 1 medicina-62-00023-t001:** Baseline demographic and clinical characteristics of the study population stratified by infertility etiology.

Characteristics	Female Factor (*n* = 37)	Male Factor (*n* = 43)	Idiopathic (*n* = 20)
Age (years), mean	36.4	34.8	36.5
BMI (kg/m^2^), mean	24.99	23.50	23.68
Smoking, *n* (%)	6 (16%)	12 (28%)	5 (25%)
AMH (mean ± SD)	3.20 ± 3.13 ng/mL	2.31 ± 0.94 ng/mL	2.55 ± 2.13 ng/mL
Total oocytes retrieved (mean ± SD)	8.68 ± 3.22	9.77 ± 2.87	9.50 ± 3.07
MII oocytes, mean	6.65 ± 2.68	7.37 ± 2.38	7.20 ± 2.6
Clinical pregnancy, (%)	32 (40.6%)	22 (53.7%)	8 (44.4%)

No differences were observed in oocyte yield or pregnancy rates across MEDAS adherence levels (all *p* > 0.05).

**Table 2 medicina-62-00023-t002:** Baseline demographic, lifestyle and dietary characteristics of the study population (*n* = 100).

Characteristics	Value
Age (years), mean ± SD	35.7 ± 3.8
BMI (kg/m^2^), mean ± SD	24.1 ± 4.7
BMI categories, *n* (%)	Underweight 7 (7%)
	Normal weight 56 (56%);
	Overweight 23 (23%);
	Obese 14 (14%)
Smoking, *n* (%)	Smokers 23 (23%)
Cause of infertility, *n* (%)	Female factor 37 (37%);
	Male factor 43 (43%);
	Idiopathic 20 (20%)
MEDAS score, mean ± SD (range)	7.6 ± 1.2 (4–11)
MEDAS adherence categories, *n* (%)	Low (0–5) 4 (4%);
	Moderate (6–9) 93 (93%);
	High (≥10) 3 (3%)
AMH (ng/mL), mean ± SD	2.69 ± 2.23

**Table 3 medicina-62-00023-t003:** Mediterranean Diet Adherence Screener (MEDAS): Items and Scoring [9].

Items	Scoring	
Use of olive oil as main culinary fat	Yes	0–1
2. Olive oil consumption per day	≥4 tablespoons/day	0–1
3. Vegetable servings per day	≥2 servings/day	0–1
4. Fruit servings per day	≥3 servings/day	0–1
5. Red/processed meat servings per day	<1 serving/day	0–1
6. Butter/margarine/cream per day	<1 serving/day	0–1
7. Sugar-sweetened beverages per day	<1 serving/day	0–1
8. Wine consumption per week	1–7 glasses/week	0–1
9. Legume servings per week	≥3 servings/week	0–1
10. Fish/seafood servings per week	≥3 servings/week	0–1
11. Commercial sweets/pastries	<3 times/week	0–1
12. Nut servings per week	≥3 servings/week	0–1
13. Preferential consumptions of poultry over red/process meats	Yes	0–1
14. Dishes prepared with sofrito	≥2 times/week	0–1

## Data Availability

Data supporting the findings of the study are available from the corresponding author upon reasonable request.

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
