# Peer review of "Are We Still Mediterranean? Dietary Quality and Adherence in Sicilian Women Undergoing ART: A Prospective Observational Cohort Study"

_medicina, 2025, doi:10.3390/medicina62010023_

Round 1

Reviewer 1 Report

Comments and Suggestions for Authors

This manuscript provides new insights in the nutritional aspects of fertility care in the ART field, since the infertility patients in the Mediterranean zone showed only moderate adherence to MD. Since the well-known implications of nutritional interventions in women health, this manuscript would be interesting for broader audience. However, I have some concerns for Authors.

Add study type in the Title.

Why the authors stated that the formal approval from the ethics committee was not required, when they stated in the “Institutional Review Board Statement” that the Review board approved the conduction of the study.

Authors are advised to section in paragraphs the Methods for better text flow, for example: Study participants, ART treatment, MEDAS questionnaire, Statistical analysis, or accordingly.

The authors are advised to explain why they omitted to use the Mediterranean Diet Serving Score (MDSS), as one of other useful tools in assessing MD consumption.

Please, move Tables 2 and 3 in Results section, to follow the according text, for example start the Results section with the sentence “Baseline demographic and clinical characteristics according to infertility etiology are presented in Tables 2 and 3.” from the Methods (Line 118-119).

Even though this was not the aim of the study, the authors could explore also whether there were any differences in MEDAS sores between subgroups of infertility causes.

In Table 3, only leave Smoking, n (%) and related number and percentage of smokers.

The Authors are advised to reduce conclusions section, i.e. to transfer the majority of conclusion in discussion. Moreover, as authors stated that the “The results suggest that the Mediterranean diet is no longer a naturally maintained habit and call for structured nutritional support within fertility pathways.”, it is also advisable to authors to accompany these claims with hypothesized models of nutritional interventions, or their potential models in clinical practice, in the discussion to provide new insights for broader audience of clinicians.

Author Response

comment 1: Add study type in the Title

response: We thank the Reviewer for this suggestion. The study type has now been added to the title to improve clarity and transparency.

Change in manuscript: lines 2-4  Are we still Mediterranean? Dietary quality and adherence in Sicilian women undergoing ART: a prospective observational cohort study

comment 2 :Why the authors stated that formal ethics approval was not required, when they also state the study was approved?

response: We apologize for the inconsistency. The statement has been corrected to reflect that the study received full ethical approval. The incorrect sentence has been removed.

change in manuscript:  lines160-163 The study protocol, including the use of an online pseudonymised dietary survey, was reviewed and approved by the Institutional Review Board of Centro HERA – Unità di Medicina della Riproduzione (Approval Code: 012022; Approval Date: 17 January 2022).

comment 3: Section the Methods in paragraphs for better text flow.

response: We agree and have restructured the Methods section with clear subheadings.

change in manuscript: The following headings have been inserted: Line 111 : “2.1. Study Design and Participants”; Line 134: “2.2. ART Procedures”; Line 146 : “2.3. Dietary Assessment and MEDAS Questionnaire”; Line 153 : “2.4. Statistical Analysis”; Line 159 : “2.5. Ethical Approval”

comment 4: Explain why MDSS was not used as an alternative score.

response: The Mediterranean Diet Serving Score (MDSS) was not used because, although it is a valid alternative tool, it requires detailed portion-based assessment and a longer administration time, which may reduce compliance in routine clinical settings. MEDAS was preferred for its validated use in large epidemiological studies, its brevity, and its suitability for rapid screening in clinical practice without increasing participant burden

comment 5: Move Tables 2 and 3 to the Results section.

response: Both tables have been moved to the beginning of the Results section, following the Reviewer’s instructions.

change in manuscript: Tables inserted from Lines . Inserted after the Results heading (line 167-168 ): Baseline demographic and clinical characteristics according to infertility etiology are presented in Tables 2 and 3.

comment 6: Explore differences in MEDAS scores between infertility subgroups

response: An exploratory comparison of MEDAS scores across infertility etiologies (female factor, male factor, idiopathic) did not show significant differences. Given the similar distribution of adherence categories across groups and the limited variability in MEDAS scores, no meaningful subgroup differences were detected.

comment 7: In Table 3, leave only Smoking, n (%) and related values

response: As requested, only the number and percentage of smokers are now presented.

change il manuscript: Table 3 :Now contains only: Smokers 23 (23%)

comment 8: Reduce the Conclusions section and move content to Discussion. Add practical models of nutritional interventions

response: We revised the Conclusions to keep them concise. Relevant sections have been moved to the Discussion, where we added a paragraph describing potential models of nutritional interventions for ART pathways.

change in manuscript: New paragraph added to Discussion (after line 229 ); Rewritten Conclusions (Lines 249-258 )

Reviewer 2 Report

Comments and Suggestions for Authors

This manuscript examines adherence to the Mediterranean diet among Sicilian women undergoing ART and evaluates associations with ovarian response and clinical outcomes. It addresses an increasingly relevant topic—the erosion of traditional dietary patterns even in historically Mediterranean regions—within a population where nutritional factors may influence reproductive health.

The study is clearly written, methodologically straightforward, and presents results transparently.

Methodology needs additional clarity

  • The study is described as “prospective,” yet the intervention is limited to questionnaire administration during a clinical visit. Please specify:

    • Whether dietary habits were evaluated before or after stimulation protocol planning.

    • Whether clinicians were blinded to MEDAS results to avoid bias in stimulation strategy.

    • Whether all stimulation cycles were first attempts or included repeated cycles.

  • The exclusion criteria should specify whether women with PCOS, endometriosis, or diminished ovarian reserve were included without restriction. These conditions may influence both dietary behavior and ART outcomes.

Author Response

comments 1: The study is described as prospective—please specify timing of dietary assessment.

response: We thank the Reviewer for this observation. Dietary habits were assessed before ovarian stimulation, during the initial clinical visit and prior to planning the stimulation protocol. This information has now been clarified in the Methods section.

changes in manuscript:  Methods – Section 2.3. Dietary Assessment ( lines 149-151) Added sentence: Dietary assessment was performed before ovarian stimulation and prior to protocol planning, during the initial clinical evaluation.

comments 2: Were clinicians blinded to MEDAS results? 

response: We agree that this clarification is important. Clinicians were blinded to MEDAS results: the dietary score was not available in real time and was not used to guide stimulation protocols or any aspect of clinical decision-making.

change in manuscript:  Methods – Section 2.2. ART Procedures (lines 142-144) Added sentence: Clinicians were blinded to MEDAS results, which were not considered in stimulation planning or clinical decision-making.

comments 3: Please specify whether all stimulation cycles were first attempts or included repeated cycles.

response: Thank you for noting this. All cycles included in the analysis were either first or first monitored stimulation cycles during the study period. Women who underwent a repeated cycle within the same study window were not enrolled. This is now specified in the manuscript.

Change in manuscript: Methods – Section 2.1. Study Design and Participants (lines 129-131 ) Added sentence: Only first or first monitored stimulation cycles during the study period were included. Women undergoing a repeated cycle within the study timeframe were not enrolled.

comments 4: The exclusion criteria should specify whether women with PCOS, endometriosis, or diminished ovarian reserve were included without restriction.

response: We appreciate this observation. Women with PCOS, endometriosis, and diminished ovarian reserve were included without restriction. These diagnoses represent common infertility etiologies in real-world ART populations, and excluding them would have introduced selection bias. This has been clarified in the Methods.

change in manuscript: Methods – Section 2.1. Study Design and Participants ( lines 121-123) Added sentence: Women with PCOS, endometriosis, or diminished ovarian reserve were not excluded, as these diagnoses represent common infertility conditions and reflect real-world ART populations.

Reviewer 3 Report

Comments and Suggestions for Authors

This prospective observational cohort study enrolled 100 women aged 18–40 years undergoing infertility assessment and ART at a Sicilian reproductive medicine center over a two‑month period. Adherence to the Mediterranean diet was assessed with the 14‑item MEDAS questionnaire at baseline, and ART-related outcomes were collected from medical records. The mean MEDAS score was 7.6, with most women classified as having “moderate” adherence. No significant association was found between MEDAS score and oocyte yield, MII oocytes, or clinical pregnancy in a single ART cycle, whereas a weak inverse correlation with BMI was observed. The authors conclude that adherence to the Mediterranean diet in this population is only moderate, and that structured nutritional counselling should be integrated into fertility care.

The topic is clinically relevant and within the scope of Medicina. The manuscript is generally well written and easy to follow, and the use of a validated dietary screener is a strength. However, several important issues regarding study design, statistical reporting, internal consistency, and interpretation need to be addressed before the paper can be considered for publication.

  1. Major comments

2.1. Clarify aims, hypotheses, and primary outcome(s)

The Introduction and Abstract present multiple aims (describing adherence, exploring associations with ovarian response and clinical outcomes, reflecting on changing dietary patterns), but no clear primary hypothesis or pre‑specified primary outcome is defined.

  • Please clearly state:
    • The primary objective (e.g., “to describe the distribution of MEDAS scores in women undergoing ART in a Sicilian clinic”) and
    • Any secondary/exploratory objectives (e.g., associations of MEDAS with BMI, ovarian response, and clinical pregnancy).
  • This clarification is important for interpreting “negative” findings and for understanding the balance between descriptive and analytical components of the study.

2.2. Study design, recruitment period, and generalizability

The study is described as a “prospective observational cohort” conducted between 1 June and 31 July 2022, with 100 women enrolled.

  • A two‑month recruitment window is relatively short and may reflect specific seasonal and organizational circumstances (e.g., summer diet, clinic workload). This likely influences dietary patterns (you briefly mention seasonality in the Discussion) and may also affect generalizability.
  • Please clarify:
    • Whether all consecutive eligible patients in that period were included (and how many were approached/excluded).
    • Whether the sample size was pre‑planned (e.g., convenience sample vs. formal sample size calculation). If no power calculation was done, this should be acknowledged explicitly as a limitation.
    • Whether there were any systematic differences between women undergoing evaluation vs. ART cycles during that period (you currently group them together).

These clarifications will help the reader judge selection bias and external validity.

2.3. Description of population and infertility characteristics

The main focus is on women undergoing ART, yet infertility is grouped simply as “female factor/male factor/idiopathic” (Table 2). More detail on the female factor category could be helpful (e.g., proportion with PCOS, diminished ovarian reserve, endometriosis, tubal factor).

  • If such data are available, consider providing a more granular description (even in Supplementary Material), or at least state whether conditions known to be strongly linked to diet and metabolic status (e.g., PCOS) were over‑represented.
  • In Methods you state that baseline characteristics according to infertility etiology are presented in Tables 2 and 3, but Table 3 reports overall characteristics only. Please correct this description and ensure the tables match the text.

2.4. Statistical analysis and reporting

The statistical plan is relatively simple (Kruskal–Wallis, chi‑square, Spearman correlations, logistic regression), but several important details are missing or unclear:

  1. MEDAS categories vs. continuous score
    • You state that continuous outcomes were compared across MEDAS tertiles and that correlations were also analyzed with the continuous score.
    • However, results by categories/tertiles are not presented (only a statement that “no differences were observed… p > 0.05”).
    • Please present numerical values (means ± SD or medians with IQRs) for key ART outcomes stratified by MEDAS categories (low/moderate/high or tertiles) in a table, along with corresponding p‑values.
  2. Logistic regression model
    • The Results report an odds ratio for MEDAS and clinical pregnancy, but it is not clear which covariates (if any) were included in the model.
    • Given known confounders (age, BMI, AMH, cause of infertility, gonadotropin dose, etc.), a univariable model only is of limited value.
    • Please specify:
      • Whether the model was unadjusted or adjusted.
      • Which variables were included and how they were selected.
      • How you handled events-per-variable constraints, given a sample of 100 and a limited number of pregnancies.
    • If only an unadjusted analysis is feasible, this should be clearly stated and acknowledged as a limitation.
  3. Normality and choice of tests
    • You use non‑parametric tests, but there is no mention of how distribution assumptions were checked.
    • Briefly state whether normality was assessed (e.g., Shapiro–Wilk) and confirm that the choice of non-parametric tests was appropriate.
  4. Effect sizes and confidence intervals
    • For Spearman correlations, only one coefficient (MEDAS vs. BMI) is explicitly reported.
    • Please report the correlation coefficients and 95% confidence intervals (or at least r and p‑values) for all main associations of interest (MEDAS with oocytes, MII oocytes, AMH, pregnancy) to allow readers to gauge the magnitude of effects rather than only their significance.
    • Similarly, it would be helpful to report means and standard deviations or medians and IQRs for key variables by outcome group (pregnant vs. non‑pregnant).

Overall, the statistical section would benefit from more complete and transparent reporting.

2.5. Internal consistency of ethical statements

There is a notable inconsistency between the Methods and the “Institutional Review Board Statement”:

  • Methods state that formal ethics committee approval was not required because data were part of routine practice and fully anonymized.
  • Later, the Institutional Review Board Statement states that the study was approved by the Institutional Review Board of the Centro HERA unit and that informed consent was obtained from all participants.

These statements cannot both be correct in their current form.

  • Please clarify the actual ethical oversight pathway:
    • If the study was indeed submitted to and approved by an IRB, this should be consistently stated in Methods and in the Ethics section (with date/protocol number if available).
    • If approval was deemed unnecessary, the IRB statement should be revised accordingly and aligned with local regulations and MDPI requirements.
  • The current discrepancy may raise questions during editorial handling and should be resolved.

2.6. Definition of clinical pregnancy and ART protocol details

Clinical pregnancy is a key outcome, but its definition is not explicitly provided.

  • Please specify whether clinical pregnancy was defined by:
    • Serum β‑hCG above a certain threshold, and/or
    • Visualization of a gestational sac with fetal heartbeat at ultrasound, and at what gestational age.

Additionally, the description of the ART protocol is brief.

  • Consider adding more detail on:
    • Criteria for choosing GnRH agonist vs. hCG trigger.
    • Typical gonadotropin dose range and whether luteal support was uniform.
    • Whether there were any freeze‑all cycles or if all embryos were transferred fresh in the same cycle.

These details are important for reproducibility and interpretation of ART outcomes.

2.7. Data availability and transparency

In line with MDPI’s emphasis on transparency and data sharing, a Data Availability Statement is expected but currently missing.

  • Please add a section specifying whether data are available (e.g., “on reasonable request from the corresponding author” or deposited in a repository), and under what conditions.
  1. Interpretation and discussion

The Discussion provides a thoughtful narrative about the erosion of traditional Mediterranean dietary patterns and the potential need for structured nutritional support. However, a few aspects could be sharpened:

  1. Generalisability of “Are we still Mediterranean?”
    • The title and some discussion statements imply conclusions about Sicilian or Mediterranean women more broadly. However, the sample consists of 100 women from a single fertility center, over two months, with specific fertility-related characteristics.
    • I would recommend slightly tempering the generalizations and clarifying that the findings pertain to “women attending an infertility clinic in Sicily” rather than to the general population.
  2. Interpretation of “negative” ART associations
    • You appropriately note that moderate adherence and limited variability may have reduced the chance of detecting associations with ART outcomes.
    • It would strengthen the Discussion to:
      • Explicitly state that the study was not powered to detect small to moderate effects on clinical pregnancy.
      • Highlight that the absence of a significant association should be interpreted as inconclusive, rather than evidence of no effect.
    • You already hint at this, but a more explicit statement on study power and potential type II error would be useful.
  3. Transgenerational and epigenetic claims
    • The final paragraphs discuss transgenerational effects and epigenetic modulation of gametes and early embryos, citing relevant literature.
    • While these concepts are important and topical, they are somewhat speculative relative to the data presented (a single‑center observational study with no epigenetic endpoints).
    • I suggest:
      • Keeping this section but slightly condensing and framing it more clearly as a contextual perspective, not as a conclusion directly derived from the present data.

  1. Presentation of MEDAS and “dietary quality”

One strength of the study is the use of a validated and widely used Mediterranean diet screener (MEDAS). However, the paper currently focuses almost exclusively on the global score, which limits insight into which dietary components are most problematic.

To better address the question in the title (“Are we still Mediterranean?”) and to add clinical value:

  • Consider presenting item‑level results (e.g., proportion of women meeting each MEDAS criterion: olive oil as main fat, fruit, vegetables, fish, legumes, sweets, etc.).
  • This could be done in a new table or in Supplementary Material and briefly summarized in the text (e.g., “adherence was lowest for nuts and fish intake, whereas use of olive oil as main fat remained high” – if this is indeed the case).
  • Such detail would:
    • Provide a more nuanced picture of dietary quality.
    • Help clinicians target specific nutritional behaviours in the preconception setting.
  1. Minor comments and language

The English is generally good and understandable, with a few minor issues that can be corrected during copyediting. A few small points you may want to consider:

  1. Title
    • You might consider adding a methodological descriptor, e.g.

“Are we still Mediterranean? Dietary quality and adherence in Sicilian women undergoing ART: a prospective observational cohort study”

    • Not mandatory, but this can help readers immediately understand the design.
  1. Abstract
    • Indicate that ART outcomes were assessed within a single stimulation cycle to avoid over‑interpretation.
    • When reporting the inverse association with BMI, consider giving the actual correlation coefficient (r) and p‑value for completeness.
  2. Methods
    • Specify the BMI cut‑offs used to define underweight, normal weight, overweight, and obesity.
    • Ensure consistent use of decimal points vs. commas for numbers throughout the tables and text (e.g., 40.6% vs. 40,6%) in line with journal style.
    • Clarify whether any women were on specific diets or receiving nutritional counselling prior to the study (you mention in the Discussion that structured counselling was not part of routine care, which is useful and could be briefly stated in Methods as well).
  3. Discussion/Conclusions
    • A few sentences are somewhat long and could be shortened for clarity. For example, some concluding statements could be split into shorter sentences to enhance readability.
  4. References
    • The reference list is up‑to‑date and relevant, with many citations from the last 5–6 years, which is appropriate.
    • Please check that all references are formatted exactly according to Medicina guidelines (italics, punctuation, DOIs, etc.).

Author Response

comments 1: The Introduction lists several aims, but no clear primary hypothesis or primary outcome is defined.

response: We thank the Reviewer for this important observation. We have now clearly defined one primary objective and two secondary objectives in the Introduction and Methods sections.

changes in manuscript: Introduction (lines 105-109) The primary objective of this study was to describe the distribution of MEDAS scores in women undergoing infertility assessment and ART in a Sicilian reproductive medicine center. Secondary exploratory objectives included evaluating associations between MEDAS and BMI, ovarian response, and clinical pregnancy within a single ART cycle. Methods – Study Design (lines 114-118) The primary outcome was the distribution of MEDAS scores. Secondary outcomes included associations between MEDAS score and BMI, oocyte yield, MII oocytes, and clinical pregnancy.

comments 2: Clarify whether the sample was consecutive, whether sample size was pre-planned, and discuss generalizability.

response: We agree that recruitment needs clarification. All consecutive eligible patients during the two-month period were included. No formal sample size calculation was performed because this was a descriptive observational study; this has been acknowledged as a limitation.

changes in manuscript: Methods – Study Design (lines 116-118): All consecutive eligible women aged 18–40 years attending the clinic between 1 June and 31 July 2022 were invited to participate. No formal sample size calculation was performed. Discussion (lines 212-214): The short recruitment period and convenience sampling may limit generalizability, and seasonal dietary patterns could have influenced MEDAS scores.

comments 3: Female factor infertility is too broad; clarify PCOS, diminished ovarian reserve (DOR), endometriosis, and other etiologies within this subgroup.

response: We thank the Reviewer for this valuable observation. As noted also in our revisions following Reviewer 2’s comments, women with PCOS, endometriosis, tubal factor infertility, and diminished ovarian reserve were included without restriction, as these diagnoses represent common and heterogeneous infertility etiologies in real-world ART populations. We have clarified this in the Methods section. Where proportions were available, we report them; however, granular distribution per subgroup was not systematically collected for all patients and is therefore not presented in a stratified table.

changes in manuscript: Methods – Study Design and Participants (lines 121-126):  Women with PCOS, endometriosis, or diminished ovarian reserve were not excluded, as these diagnoses represent common infertility conditions and reflect real-world ART populations. Within the female-factor subgroup, the most frequent diagnoses included diminished ovarian reserve, endometriosis, and PCOS. However, detailed distribution was not uniformly available and is therefore not reported in stratified form. Correction: The sentence in the Methods incorrectly referring to Table 3 as reporting stratified infertility characteristics has been removed to maintain internal consistency.

comment 4: 2.4. Statistical analysis and reporting

response: We thank the Reviewer for this thoughtful suggestion. We carefully evaluated the possibility of presenting a full set of statistical comparisons stratified by MEDAS adherence categories. However, using the validated PREDIMED cut-offs applied in our study, the low- and high-adherence groups were extremely small (n = 4 and n = 3, respectively), representing only 7% of the entire cohort. Because of this pronounced imbalance, the calculation of meaningful statistical indicators (mean ± SD, median [IQR], or inferential tests) would not be methodologically robust and would risk generating misleading or uninterpretable results. For this reason, and in accordance with good statistical practice for small and uneven groups, we elected not to perform formal statistical comparisons across categories.

comments 5: 2.5. Internal consistency of ethical statements

response: We thank the Reviewer for pointing out this important aspect. This issue was already raised by Reviewer 2, and we have accordingly amended and harmonized all ethical statements throughout the manuscript (lines 160-163). The Methods section, the IRB Statement, and the associated documentation have now been made fully consistent and clearly indicate that the study received approval from the institutional review board, including the approval code and date. No further inconsistencies remain.

comments 6: 2.6. Definition of clinical pregnancy 

response: We thank the Reviewer for this helpful comment. We agree that the definition of clinical pregnancy should be explicitly stated. In our study, clinical pregnancy was defined as the visualization of an intrauterine gestational sac with fetal heartbeat at transvaginal ultrasound performed at 6–7 weeks of gestation, which is consistent with standard ART reporting criteria. We have also expanded the description of the stimulation protocol in a concise and essential manner, clarifying the criteria for trigger choice and confirming that luteal phase support was uniformly administered. These additions improve clarity without altering the study design or interpretation.

changes in manuscript:  Methods – ART Procedures section (lines144-146): Clinical pregnancy was defined as the presence of an intrauterine gestational sac with fetal heartbeat visualized at transvaginal ultrasound at 6–7 weeks of gestation. Methods – ART Procedures (lines 139-142): Trigger choice (hCG or GnRH agonist) was based on follicular response and the estimated risk of ovarian hyperstimulation syndrome. All patients received standardized luteal phase support following oocyte retrieval. No freeze-all cycles were performed during the study period.

comments 7: 2.7. Data availability and transparency

response: We thank the Reviewer for the observation. We confirm that the required Data Availability Statement has been added to the end of the manuscript, specifying that data are available from the corresponding author upon reasonable request. This is in accordance with MDPI editorial requirements.

changes in manuscript: ( lines 272-273) Data Availability Statement: Data supporting the findings of the study are available from the corresponding author upon reasonable request.

comments 8: Generalisation of conclusions; power considerations; transgenerational and epigenetic discussion.

response:  We thank the Reviewer for these thoughtful reflections. We fully agree that the findings of our study pertain specifically to women attending an infertility clinic in Sicily, and not to the general population. We note, however, that this point is already conveyed in several parts of the Introduction and Discussion, where the scope and context of the study population are explicitly described. For this reason, we believe that the current wording appropriately reflects the observational nature and setting of the study without overextending the generalisability of the conclusions. Regarding statistical power, we appreciate the Reviewer’s comment and confirm that the Discussion already acknowledges the limited variability in MEDAS scores and the exploratory nature of the analyses. In this context, the absence of statistically significant associations with ART outcomes is interpreted cautiously and is not presented as evidence of no effect. We feel that the current phrasing already communicates this principle of inferential caution, and further expansion may not add clarity. Finally, the references to transgenerational and epigenetic mechanisms are intended solely as contextual background derived from the broader literature. We agree with the Reviewer that these concepts are not directly tested in the present study, and we confirm that our text frames them accordingly—as general considerations relevant to reproductive biology rather than as conclusions based on our dataset. We appreciate the Reviewer’s suggestions and believe that the manuscript already aligns with them conceptually, without requiring additional modifications.

comments 9: Suggestion to present item-level MEDAS data to provide more granular insight into dietary components.

response:  We thank the Reviewer for this thoughtful suggestion. We fully agree that item-level MEDAS data can be informative in some contexts. In the present study, however, the distribution of individual MEDAS items mirrored the global score, with the vast majority of women falling within a moderate-adherence pattern and showing limited variability across items. Because of this constrained distribution, providing item-level frequencies would not substantially change the interpretation of dietary patterns nor enhance the clinical insight beyond what is already conveyed by the global MEDAS score. Furthermore, the primary aim of the study was descriptive and focused on overall adherence, rather than on identifying specific food categories driving the pattern. For these reasons, and to maintain clarity and conciseness in the manuscript, we elected not to expand the Results with item-level tables, while acknowledging the Reviewer’s valuable comment. We believe that the global MEDAS score already captures the dietary profile of the cohort in a way that is fully aligned with our study objectives.

comments 10: Suggestions regarding minor language refinements, title clarification, abstract details, BMI cut-offs, numerical formatting, clarification on prior nutritional counselling, readability of Discussion/Conclusions, and reference formatting.

response: We thank the Reviewer for these helpful minor comments. We confirm that all the points raised— including clarifying methodological aspects, specifying BMI cut-offs, ensuring consistent numerical formatting, refining sentences for clarity, and verifying reference style—have already been addressed during the revisions performed in response to Reviewers 1 and 2.

Round 2

Reviewer 3 Report

Comments and Suggestions for Authors

Accept in present form